# Mimicking Mechanics: A Comparison of Meat and Meat Analogs

**DOI:** 10.3390/foods13213495

**Published:** 2024-10-31

**Authors:** Skyler R. St. Pierre, Ellen Kuhl

**Affiliations:** Department of Mechanical Engineering, Stanford University, Stanford, CA 94305, USA

**Keywords:** meat analogs, alternative protein, texture profile analysis, mechanical properties, stiffness

## Abstract

The texture of meat is one of the most important features to mimic when developing meat analogs. Both protein source and processing method impact the texture of the final product. We can distinguish three types of mechanical tests to quantify the textural differences between meat and meat analogs: puncture type, rheological torsion tests, and classical mechanical tests of tension, compression, and bending. Here, we compile the shear force and stiffness values of whole and comminuted meats and meat analogs from the two most popular tests for meat, the Warner–Bratzler shear test and the double-compression texture profile analysis. Our results suggest that, with the right fine-tuning, today’s meat analogs are well capable of mimicking the mechanics of real meat. While Warner–Bratzler shear tests and texture profile analysis provide valuable information about the tenderness and sensory perception of meat, both tests suffer from a lack of standardization, which limits cross-study comparisons. Here, we provide guidelines to standardize meat testing and report meat stiffness as the single most informative mechanical parameter. Collecting big standardized data and sharing them with the community at large could empower researchers to harness the power of generative artificial intelligence to inform the systematic development of meat analogs with desired mechanical properties and functions, taste, and sensory perception.

## 1. Introduction

The protein sources we choose to eat can have a large impact on the environment, global warming, and our own health [1,2]. Animal proteins from commercial farming are estimated to require 2.4–33x more water and land and produce 2.4–240x more global gas emissions than plant proteins [3]. Yet, high-protein plants like soy are frequently criticized for not being more sustainable than animal meat due to their high land use, which can put a strain on the local ecosystem [4,5]. In addition to plant proteins, researchers are now beginning to look towards other protein sources such as algae [4], fungi [6], and cultured meat [7] to provide more sustainable alternatives to animal proteins.

In consumer surveys, meat is often associated with *“delicious”*, while meat analogs are associated with *“disgusting”* [8]. Mimicking the sensory experience of biting down and chewing meat is critical for meat analogs to sway consumers to change their grocery shopping habits [8,9,10]. While sensory panels and consumer surveys provide valuable information about the perception of texture and taste, ultimately, these are subjective measures [11]. Mechanical testing, in contrast, provides quantitative values to directly compare meat analogs and meats [12].

Meat analogs are capable of mimicking the mechanics of meats with the right fine-tuning of their protein sources, compositions, and processing methods. While various methods exist to probe the mechanics of meat, selecting the best mechanical test, protocol, and data analysis is not a standardized process, making it hard to compare results across different studies. For example, a Warner–Bratzler shear test measures the shear force required to cut through a piece of meat and a mechanical compression test measures the meat stiffness. From these tests, we know that commercially available *comminuted meat* analogs already have Warner–Bratzler shear force and stiffness values similar to many meat products. At the same time, *whole meat* products remain more challenging to replicate in terms of Warner–Bratzler shear force and anisotropy. This raises the question whether and how we can standardize meat testing, systematically collect standardized parameters across all tests, and share them with the community at large to inform the development of meat analogs with targeted mechanical properties and sensory perception.

*Here, we review the most popular testing methods for meat, discuss their advantages and limitations, and make recommendations for improvement, with the goal to standardize meat testing and enable rigorous cross-study comparisons.* Throughout this review, we use the term *meat analog* to refer to a product engineered from alternative protein sources, plant, fungal, algal, milk, or egg protein, or from lab grown cells designed to mimic whole or comminuted meat. Since both *protein sources* and *processing methods* contribute to the mechanical properties of the final product, we begin with a brief overview of the protein sources and processing methods that are either already commercially used or currently in development. For the first time in the literature, we then present compiled cross-study values of the *Warner–Bratzler shear force* and *stiffness* of various whole and comminuted meats and meat analogs. Our study suggests that, with the right fine-tuning, meat analogs are well on the way to accurately mimicking the mechanical signature of various types of meat.

## 2. Making Meat Analogs

### 2.1. Protein Sources

Figure 1 and Table 1 summarize thirty alternative protein sources with their protein type and protein content. All products are sorted by dry-weight protein content ranging from 31.6% for hemp seed to 2.4% for mushroom [13]. There are various potential non-animal *protein types* for meat analogs, including legumes with beans, lentils, peas, and peanuts; oil seeds with hemp, sunflower, and flax seeds; nuts with almond, cashew, and pecan; cereals with wheat, rice, oat, barley, and corn; vegetables with potato and pea; and fungi with mushroom [14,15,16,17,18,19].

The three main commercial *protein sources* are soy, pea, and wheat protein [20]. Soy and pea protein have similar functionality with the ability to aggregate, gel, and form fibers through heating and extrusion [20]. However, structures made of pea protein are weaker than those of soy protein, but hydrocolloids can modulate their stiffness [19]. Wheat protein is often used in conjunction with a legume protein to provide a more fibrous texture and increase elasticity [20]. While soy and wheat have good functional properties for the production of meat analogs, companies and researchers are increasingly looking towards other protein types due to the allergenicity of soy and wheat and the lack of many essential amino acids in wheat protein [21]. The proteins from other legumes vary widely in their emulsion, foam stabilization, and gelling capacities, limiting their current applications; chickpea and mung bean protein have the most promising functional properties for use in meat products [19]. Fungi and algae are non-plant protein alternatives that can also be turned into meat analogs. Algae are not yet used commercially in meat products, but preliminary research shows that algal protein can create products with high moisture content leading to a juicy and soft texture [4]. Fungal proteins are able to increase both the flavor and nutritional content of meat analogs and can be processed into a fibrous structure [20]. Fungi-derived products already exist on the market in the form of deli slices, nuggets, and steaks [6].

While plant, fungal, and algal protein sources are both environmentally friendly and suitable for a vegan diet, researchers are also looking into animal-based alternative protein sources and reduced meat formulations that have the potential to closely match the taste and texture of meat with the promise of reduced environmental impacts and improved health compared to traditional meat products [22]. Milk protein and egg protein have a lower environmental impact than traditional meat due to decreased water and land use [22]. Animal–plant protein mixes are an alternative to either entirely animal- or plant-based meats but suffer from a poor physical connection between the animal- and plant-based components [22]. Animal cells can be cultured in a lab to create “in vitro meat”, which is predicted to have a lower environmental impact than traditional meat while maintaining the same taste; however, it is currently unclear how to scale up the production of cultured meat in a cost- and energy-efficient manner [2].

### 2.2. Processing Methods

In Figure 2, we have divided the processing methods to create texture from the protein sources into those used commercially and those in development. Top-down and bottom-up processing methods are capable of creating mechanical anisotropy to mimic the fiber alignment and structural organization of muscle [23]. Bottom-up methods first make fibrillar structural elements at small length scales that can then be assembled into larger scale anisotropic products, while top-down methods mimic the anisotropy of meat on the large scale without having smaller building blocks as components [23]. Bottom-up methods include fermentation of fungal proteins, cell culturing with bioreactors, wet spinning, electrospinning, and 3D printing [23,24]. Top-down methods include extrusion, the mixing of proteins and hydrocolloids, freeze structuring, and shear cell and Couette cell technology [23,24].

Figure 2 divides meat products into two main categories: *comminuted*, including minced, chopped, and ground, and *whole muscle* [31]. Currently, the only commercially used processing methods for producing whole-muscle analogs are extrusion and shear cell technology [31]; however, it remains a challenge to produce analogs with the complex structure of animal muscle, including fat, tissue, and fiber orientation [32]. Extrusion is divided into low-moisture extrusion, which creates a dry textured vegetable protein that must be rehydrated before consumption, and high-moisture extrusion, which results in meat products with increased springiness and cutting strength compared to low-moisture extrusion [33]. Extrusion methods are not easily tunable due to the large number of process parameters throughout the extrusion device making it hard to know which parameters to change for certain desired output qualities [33]. Additionally, extrusion outputs a dense, impermeable product that cannot both keep fiber orientation and be shaped into whole-muscle cuts of a realistic size [32]. In contrast, shear cell technology has highly controllable parameters, shear rate, and temperature, which form different structures depending on their settings [34]. Yet, shear cell technology is also unable to provide a texture that truly mimics beef [32].

Comminuted meat analogs are much simpler to produce than whole-muscle analogs as they do not need to have large length-scale anisotropy [35]. Processing methods such as extrusion, mixtures of proteins and hydrocolloids, and fermentation of fungal protein are used commercially to create comminuted meats [20,25]. Other methods such as 3D printing, electrospinning, wet spinning, freeze structuring, and cell culturing are not yet in commercial use but are increasingly used in research settings to create comminuted meat analogs [19,24,25]. Ultimately, for both comminuted and whole-muscle analogs, the final texture is the combined result of the protein source, non-protein ingredients, pH and salts, temperature, and degree of shear induced by the processing method [36].

## 3. Evaluating the Mechanical Properties of Meat and Meat Analogs

### 3.1. Eating Biomechanics

The first step in creating realistic meat analogs is to understand oral processing or how people eat, from the first bite, via chewing and mixing with saliva, to form a bolus and swallowing [37,38]. Simultaneously, sensory information from sight, taste, smell, sound, and feel influence our perceptions of taste [37,38]. Tough, dry meat requires more lubrication and more time chewing before it is swallowed [39]. Interestingly, the rate of chewing remains constant regardless of meat tenderness [40]. Elderly people apply less chewing force and chew for longer compared to young people [41]. Understanding how people chew meat is essential to designing mechanical tests that capture the sensory experience of different aspects of the chewing process.

### 3.2. Mechanical Tests

Several types of mechanical tests have been proposed to quantify the differences people may feel when biting and chewing meat [37]. In Figure 3, we have divided these tests into three categories: *puncture-type* tests, where the meat sample is cut by one or several blades; *rheological* tests, where the sample undergoes shear due to torsional rotation; and *classical mechanical* tests of uniaxial tension, compression, and bending [42]. The raw instrument data recorded are force or torque and displacement or torsion angle over time.

From the raw data, we can derive key mechanical properties depending on the type of test. These properties are used to directly compare different attributes of meats [43]. For puncture-type tests, the peak force to shear through the sample is taken as the max recorded force with units of Newtons or kilograms of force [44]. Kilograms can be converted to Newtons by multiplying the value by 9.80665 m/$s^{2}$, standard gravity. The total work to shear the sample is calculated from the area under the force-displacement curve with units of N · m. The Warner–Bratzler shear test with a triangular blade opening, Kramer shear cell with multiple flat blades, a single razor blade, and needles are all used to perform puncture-type tests on meat [45,46].

**Figure 3 foods-13-03495-f003:**
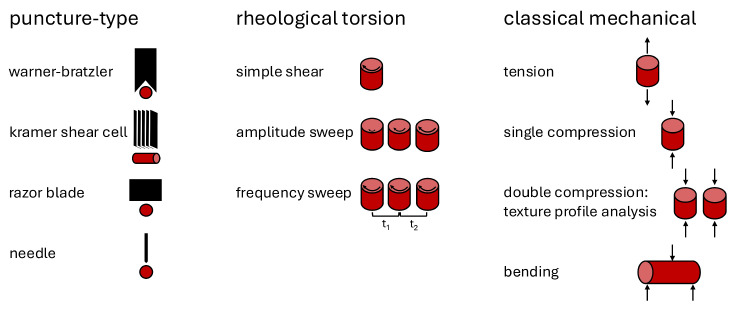
Mechanical tests of meats and meat analogs can be classified into puncture-type tests, left, rheological torsion tests, middle, and classical mechanical tests, right. Puncture-type tests include the Warner–Bratzler shear test, the Kramer shear cell, razor blades, or penetrating needles [45,46]. Rheological torsion tests include simple shear, amplitude sweeps, and frequency sweeps [43,47]. Classical mechanical tests include tension, single compression, double compression (also known as texture profile analysis test), and bending [43,47,48,49].

Rheological tests can quantify the viscoelastic, shear, and shear-rate behavior of food [47]. Meat is a viscoelastic material as a result of its complex material structure, including muscle fibers, fat, tendons, and blood [32,50]. Fluids like honey are primarily *viscous* materials for which the stress depends purely on the deformation rate such that $\sigma =\eta \cdot \dot{\gamma }$, where σ is the stress, η is the viscosity, and $\dot{\gamma }$ is the shear rate [51]. Solids like cheese are primarily *elastic* materials for which the stress depends on the deformation itself such that σ=μ·γ, where σ is the stress, μ is the material shear modulus, and γ is the amount of shear [52]. Amplitude or frequency oscillation sweep tests are dynamical mechanical tests to determine the storage modulus, loss modulus, complex modulus, and phase shift tanδ [53]. The storage modulus, G’, measures the elastic, solid-like behavior, while the loss modulus, G”, measures the viscous, fluid-like behavior. The complex modulus, G*, describes the viscoelastic behavior, while tanδ is the phase-shift between the storage and loss moduli [53]. Rheometers can also quantify the simple shear behavior with a linear assumption between torque, torsion angle, shear, and shear stress [43].

Classical mechanical tests like uniaxial tension and compression and three-point bending have also been used extensively to characterize meat products [42]. For uniaxial tension and compression, the elastic modulus or Young’s modulus, *E*, which represents the stiffness of the material, is simply the slope of the linear part of the stress–strain curve after the initial toe region or pre-load such that σ=E·ε, where σ is the stress, *E* is the elastic modulus, and ε is the amount of strain. For viscoelastic materials, the elastic modulus depends not only on the relative deformation but also on the rate of deformation [54]. The yield stress, the point at which the curve becomes non-linear; the ultimate strength, the maximum stress reached; and the toughness, the area under the stress–strain curve, are three mechanical characteristics that describe the inelastic behavior [55]. In addition, if cameras are used to capture the sample’s three-dimensional behavior during testing, the Poisson’s ratio can be calculated as the negative ratio between the transverse strain and the axial strain in the loading direction [55]. Creep tests and stress relaxation tests can measure the viscoelasticity of the sample [56,57]. Bending tests return similar metrics using similar derivations to those from uniaxial tension and compression, but these metrics, such as the flexural modulus, are specific to the geometry of the test [49]. The ISO, International Standard for Organization, provides guidelines for how to perform these mechanical tests. These guidelines have initially been designed for metals and plastics, so they are often insufficient to uniquely define the mechanical testing of soft matter like meat because of its non-standard shape and a complex mechanical behavior [58]. This explains, at least in part, why parameters like the rate of displacement, pre-load, and sample dimensions vary widely between research groups, which makes cross-study comparisons difficult [59].

#### 3.2.1. Warner–Bratzler Shear Force

The Warner–Bratzler shear test is a popular test in the meat industry that is strongly correlated with the perception of tenderness [42,45,60]. It uses a specific blade with a triangle opening, the Warner–Bratzler blade, to cut through the sample. The maximum force during the cut is the Warner–Bratzler shear force. While this test is simple to perform, its results may vary greatly because of variations in the test setup, including samples of different shapes, square or round, samples with different cross-sectional areas, different cutting rates, different blade thicknesses, and different blade opening angles [60,61,62]. While the most commonly accepted Warner–Bratzler shear test requires that the sample is cut completely through [61], some researchers cut the sample only partially [63]. Lastly, samples with a fiber direction may be cut either longitudinally or transversely; ideally, both shear forces should be reported, but if sample sizes are limited, transverse shear forces are preferable [61]. All of these possible variations impact the resulting shear force and may explain the large variations in the reported Warner–Bratzler shear force values.

Figure 4 summarizes the reported Warner–Bratzler shear force values across several studies for a wide selection of whole-muscle and comminuted meats and meat analogs. Here, we only included results where the test was specified as the Warner–Bratzler shear test and excluded other puncture-type tests. The products are arranged from the maximum reported Warner–Bratzler shear force to the minimum, from left to right. When a range of values is reported, depending on factors such as production type, sample to sample variation, different compositions of the meats, or cooking time, the bar graphs report the whole range. Whole muscle meat references include raw steak [64,65], cooked chicken breast [16,65,66], cooked steak [64,65], microalgae and soy extrude [16], raw pork [65], cooked pork [65], hemp and soy extrude [65], raw chicken breast [65], hemp and mungbean extrude [65], cooked ham [67], pumpkin and mungbean extrude [65], hemp extrude [65], pumpkin and soy extrude [65], pumpkin extrude [65]. Comminuted meat references include pork salami [68], pork sausage [69], blended beef patty [70], quail sausage [71], lamb sausage [72], veggie burger [73], chicken sausage [63], chicken patty [74], plant sausage with psyllium [75], reduced meat sausage [63], cooked beef patty [76], meat-free sausage [63], minced pork steak [77], textured soy protein patty [76], minced hybrid steak [77], tofu [16], cooked pork patty [76].

In general, whole-muscle meats have higher Warner–Bratzler shear force values than comminuted meats, for both meats and meat analogs. Comminuted meat analogs are well within the range of comminuted meats, while whole-muscle meat analogs tend to be on the lower range of whole-muscle meats. Yet, with the right combination of protein sources and fine-tuned processing parameters, meat analogs have the potential to mimic whole-muscle meats like steak, chicken, and pork.

#### 3.2.2. Texture Profile Analysis

Texture profile analysis is another popular method to characterize the mechanical properties of meats which are well correlated to sensory tests [37,78]. It consists of a double compression test at rates approximating chewing speed [37,78]. The double compression tests are performed at a fixed amount of compression, typically between ε=−20% and ε=−80% peak strain. We can directly determine hardness, springiness, adhesiveness, cohesiveness, brittleness, gumminess, and chewiness from the recorded force versus time and deformation versus time curves [37,47,78]. These parameters are derived from the peak forces of the first and second loading cycles $F_{1}$ and $F_{2}$, the associated loading times $t_{1}$ and $t_{2}$, the areas under the loading paths $A_{1}$ and $A_{3}$, the areas under the unloading paths as $A_{3}$ and $A_{4}$, and the peak stress $\sigma_{1}=F_{1}/A$, where *A* denotes the specimen cross-section area [48]. We have summarized the most common calculations for the main parameters in Table 2. While easy to perform, this test method suffers from lack of standardization, inconsistent definitions of the output parameters, and user confusion over mutually exclusive parameters like chewiness for solid foods and gumminess for semi-solid foods [37,78,79].

Texture profile analysis is typically performed in a rheometer with parallel Peltier plates, as shown in Figure 5 [47]. A circular punch is used to create samples of a known diameter, and the user must select the amount of compression and the rate to compress the samples [47]. Sample cross-sectional area, amount of compression and compression rate vary hugely between research groups, making cross-study comparisons difficult. In the case of meat, standardizing cooking time and temperature is another important element to consider, although the effect of cooking on texture profile analysis parameters for meat analogs is not yet clear, with one study reporting nearly equal changes in parameter values for meat analog patties as meat patties [49], while another study found greater changes for the meat patties [80]. Additionally, the precise definition of the textural profile analysis parameters is not unique. Springiness, for example, is defined in three different ways, as a distance with units of mm [81], a ratio with no units [71], or the second peak force with units of N [64]. Springiness is used to calculate chewiness as hardness [N] times cohesiveness [-] times springiness, so chewiness would have units of N mm, N, or N^2^, respectively, depending on the definition of springiness. Hardness, the peak of the first compressive force, with units of N, is meaningless to compare across studies as it depends on rate, sample area, and amount of compression [79]. Finally, not all parameters are reported in every study, and without the publication of the raw data, it is often impossible for other researchers to calculate unreported values.

### 3.3. Stiffness

Motivated by the limitations of the texture profile analysis test, we propose stiffness as the single best metric to enable comparison across different products and studies [43]. Stiffness, the slope of the linear region of the stress–strain curve, is directly calculated from force and deformation, scaled by sample area and height. This eliminates three of the four sources of variability from the texture profile analysis. However, since most meats and meat analogs are not purely elastic, stiffness measurements may depend strongly on the testing rate and can vary significantly for different rates [54]. Humans chew at a rate estimated to be between 33 and 66 mm/s [78], and most studies mimic rates in this range for their compression tests. However, using strain rate in mm/s means that the time to compress to a desired amount will change depending on the height of the sample. For example, at a rate of 1 mm/s, a 1 mm thick sample will take only 0.5 s to compress to half its size, while a 10 mm thick sample will take 5 s, which seems unrealistically long.

To standardize compression tests, the rate of deformation should be tied to the estimated time to chew a sample. One study of grilled meat found that people take on average 0.75 s per chew [82], while another study found that whole meat takes 0.27 s per chew, compared to 0.18 and 0.23 s per chew for comminuted meat and restructured beef [83]. In studies looking at the effect of sample thickness on mastication patterns, increasing sample thickness results in increased bite duration but not in a clear one-to-one relation [84,85]. For example, one study on carrot samples found that bite duration increases by 167%, from 0.409 to 0.683 s, for a sample thickness increase of 800%, from 2 mm to 16 mm [84]. Taken together, we strongly recommend to prescribe and report the strain rate in %/s to control the time for each chew relative to the amount of desired compression, independently of the sample height.

Figure 6 compiles reported stiffness values for meats and meat analogs from either single compression tests or double compression texture profile analysis. As we have discussed, this comparison is limited by the different rates of testing in different studies. Yet, it provides a first glance at the different stiffnesses between whole, comminuted, animal-based, and plant-based meats across multiple studies. Whole-muscle meat references include raw chicken breast [12,47], pea extrude [86], cooked chicken breast [66], soy extrude [86], chick’n [12], cooked steak [87]. Comminuted meat references include cooked beef burger [49], cooked tesco burger [49], potato protein alginate [88], tufurky deli slice [43], cultured meat sausage [47], tofurky roast [12], cooked beyond burger [49], canned ham [89], podwawelska sausage [89], raw beef burger [49], sausage [12], plant sausage [12], turkey deli slice [47], spam turkey [12], turkey sausage [12], hotdog [12], plant hotdog [12], firm tofu [12], raw tesco burger [49], extra firm tofu [12], raw beyond burger [49].

Notably, both whole and comminuted meat analogs have comparable stiffness to meats. We observe that, in most cases, despite sample-to-sample variations, varying composition, meat content, and cooking time, the reported stiffness values all lie within the same order of magnitude, around 100–200 kPa. However, cooked plant-based burgers were approximately 2.5x less stiff than beef burgers with varying beef content [49]. Strikingly, the beyond burger and tesco burger increased in stiffness approximately 58x and 16x from raw to cooked, respectively, while the beef burger increased only 9x. The sausage- and hotdog-type products have similar stiffnesses for meats and meat analogs. Whole meats have a stiffness in the range of 5–200 kPa, while comminuted meats have a larger stiffness range of 2–1200 kPa, although most fall between 2 and 420 kPa. The wide range of stiffness values of the potato protein alginate meat analog shows the potential of plant-based meats to mimic a wide range of animal meats with appropriate fine-tuning of composition and process parameters [88].

## 4. Discussion

The protein sources selected for making meat analogs impact both the nutritional value and texture of the resulting product as Figure 1 and Figure 2 suggest. Although soy, pea, and wheat are the most frequently used protein sources today [20], proteins from other legumes like chickpea and mung bean, algi, and fungi, show promise in improving the nutrition, taste, and texture profiles of meat analogs [4,19,20]. Once the protein source is selected, the processing method strongly influences the anisotropy of the resulting product, with methods like extrusion allowing for anisotropy at large enough length scales to mimic the fibrous nature of whole-meat products [33]. In contrast, comminuted or minced meats are much easier to produce as they do not require large length-scale anisotropy [35]. Naturally, more processing methods exist commercially to make comminuted products [20]. The selection of ingredients, formula, and process are guided by mechanical tests that are specifically designed to mimic the sensory experience [37]. In this review, we have compared different testing methods and discussed whether and how we can standardize the mechanical testing to characterize meat and meat analogs.


*Mechanical Tests Suffer from a Lack of Standardization*


We have divided the mechanical tests used to probe meat and meat analogs into puncture-type tests, rheological tests with torsional rotation, and classical mechanical tests of uniaxial compression, tension, and bending as summarized in Figure 3. Of these, the most popular tests for meat, the Warner–Bratzler shear test and texture profile analysis are relatively simple to understand and perform but suffer from a lack of standardization, which makes comparisons across different studies difficult if not impossible [37,60,62,78,79]. Parameters such as sample cross-sectional area, rate of displacement, amount of compression, and blade thickness vary across studies and impact the resulting values. Texture profile analysis also suffers from inconsistent parameter definitions and user confusion over which mutually exclusive parameters to report [37,78,79].


*Stiffness Is the Most Consistent Mechanical Parameter for Cross-Study Comparisons*


Given the wide variability in the Warner–Bratzler shear test and texture profile analysis, we propose to use sample stiffness as the most important parameter to report. Its unique and standardized definition as the initial slope of the stress–strain curve is independent of sample cross-section and height and allows for cross-study comparisons across various meats and meat analogs. Although the recorded stiffness is sensitive to the rate of loading, especially for the more viscous products, it inherently incorporates both sample cross-sectional area and amount of compression, minimizing two out of three of the biggest sources of variability in the Warner–Bratzler shear test and texture profile analysis. Additionally, we recommend to prescribe the strain rate and reported it in relative units of %/s rather than absolute units of mm/s to control the total time to deform. Prescribing strain rates independently of the sample height is critical to mimic a constant bite time instead of assuming that the bite time scales linearly with the sample height [84].


*Meat Analogs Successfully Mimic the Shear Force and Stiffness of Whole and Comminuted Meats*


As Figure 4 and Figure 6 suggest, the different compositions, production types, processing methods, and cooking times affect how well meat analogs can mimic the mechanical behavior of whole and comminuted meat products. In general, whole-muscle meats and meat analogs have higher Warner–Bratzler shear force values than comminuted products. The meat analogs we found in our literature search seemed most suited to mimic comminuted meat products, which have lower shear values. Similarly, meat analogs like sausages and hotdogs have similar stiffness values as their corresponding meat products, in the range of 420 kPa or under, but do not yet seem capable of mimicking the high stiffness of cooked beef burgers in the range of 900 kPa or higher [49]. Although both Warner–Bratzler shear force and stiffness are simple to test and provide straight-forward comparison metrics, these values are from a *single mode*. When sample quantity allows, we recommend to perform *multi-mode* testing, such as tension, compression, and shear, to probe the full 3D mechanical behavior [12,43].

## 5. Conclusions

Meat analogs are capable of mimicking the mechanics of meats with the right fine-tuning of their protein sources, compositions, and processing methods. Deciding on the right mechanical test, protocol, and analysis of the raw data is not yet a standardized process, making it hard to compare results across different studies. Here, we critically discuss the most popular testing methods for meat and rationalize that stiffness is the best metric to compare meat products using either single compression or double compression texture profile analysis testing. Commercially available comminuted meat analogs already have similar Warner–Bratzler shear forces and stiffness values as many meat products. Whole meat products remain more challenging to replicate in terms of Warner–Bratzler shear force and anisotropy. Collecting large data sets and sharing them with the community in a more standardized fashion could empower researchers to leverage generative artificial intelligence to inform the development of meat analogs with desired mechanical properties, taste, and sensory perception.

## Figures and Tables

**Figure 1 foods-13-03495-f001:**
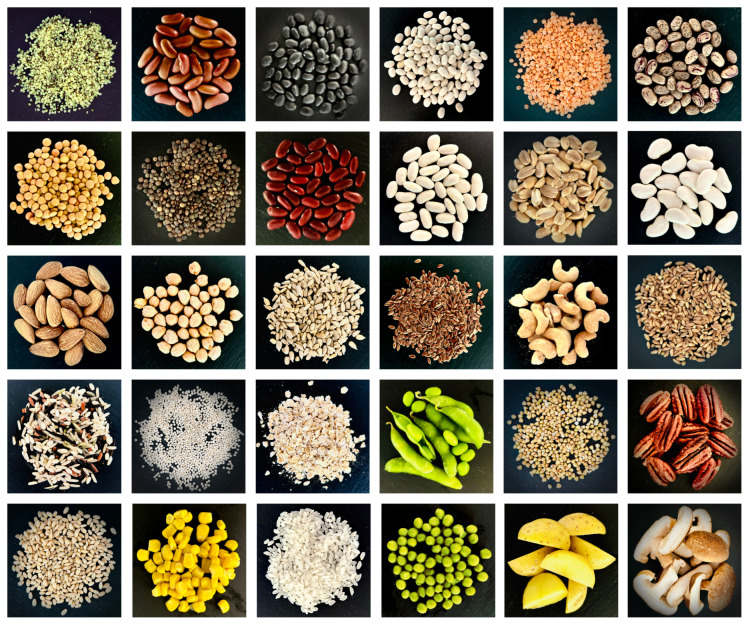
Thirty popular alternative proteins sorted by dry-weight protein content according to Table 1, including hemp seed 31.6%, kidney bean 25.9%, black bean 24.4%, navy bean 24.1%, red lentil 23.9%, pinto bean 23.7%, brown lentil 23.6%, green lentil 23.6%, small red bean 23.5%, large white bean 23.4%, peanut 23.2%, large lima bean 21.5%, almond 21.4%, chick pea 20.5%, sunflower seed 18.9%, flaxseed 18.3%, cashew 17.4%, farro 15.6%, wild rice 14.7%, quinoa 14.1%, oat 13.5%, soy bean 11.9%, buckwheat 11.1%, pecan 10.0%, barley 9.9%, corn 9.4%, white rice 7.0%, green peas 5.4%, potato 2.6%, mushroom 2.4%, from top left to bottom right.

**Figure 2 foods-13-03495-f002:**
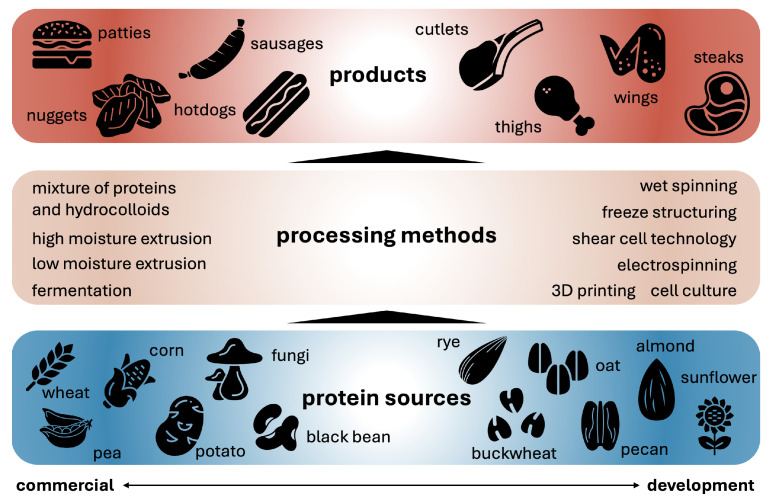
Developing meat analogs involves the selection of protein sources, bottom, processing methods, middle, and product shapes, top. Protein sources can be divided into those that are already commercially available, left, and those that are still in development, right [18,20,25,26,27,28,29]. Processing methods can be divided into commercially available, left, and in development, right [20,23,25,26,27,30]. Final products can be divided into comminuted meat analogs, left, and whole-muscle analogs, right [26,27,30].

**Figure 4 foods-13-03495-f004:**
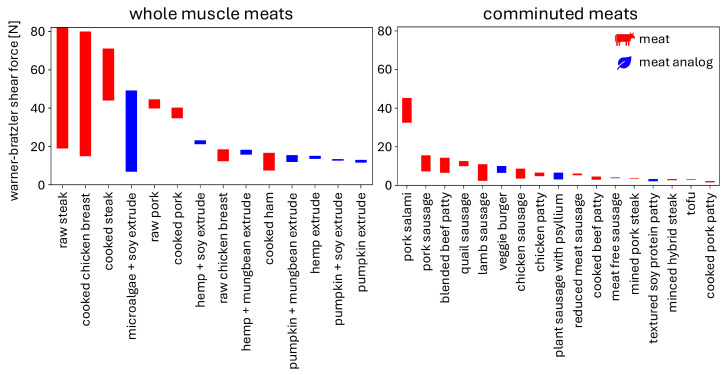
Warner-Bratzler shear forces for whole-muscle meats and for comminuted meats and meat analogs across several studies. The ranges represent different reported values across studies, sample to sample variation, different cooking times, different production methods, different compositions, and other factors.

**Figure 5 foods-13-03495-f005:**
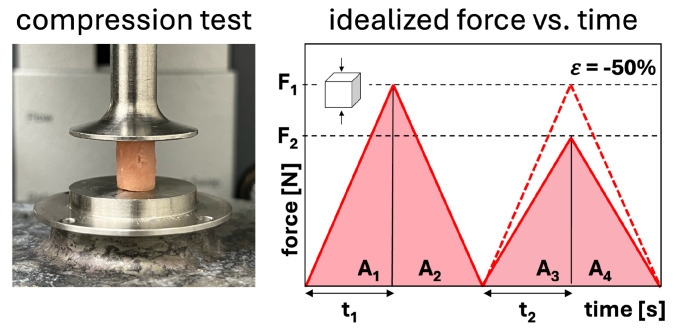
Texture profile analysis preformed on a cylindrical sample of meat, mounted between two parallel Peltier plates of the rheometer (**left**). Definition of texture profile analysis parameters from Table 2 in the idealized force versus time curves of the double compression test (**right**).

**Figure 6 foods-13-03495-f006:**
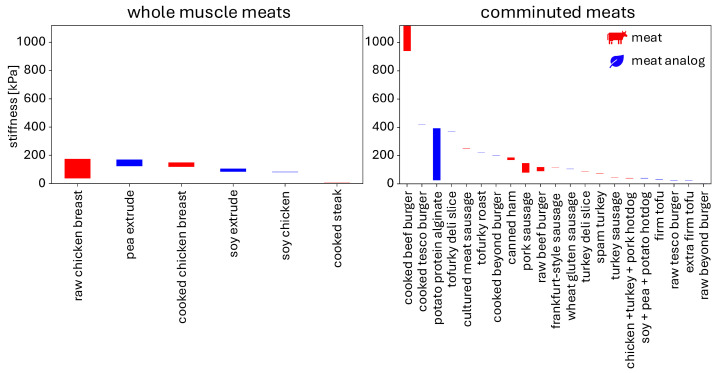
Stiffness values for whole-muscle meats and for comminuted meats and meat analogs across several studies. Stiffnesses result from single or double compression texture profile analysis tests. The ranges represent different reported values across studies, sample-to-sample variation, different cooking times, different production methods, different compositions, and other factors.

**Table 1 foods-13-03495-t001:** Thirty popular alternative proteins with their protein source, protein type, and protein content, according to Figure 1, sorted by dry-weight protein content [13].

Protein	Protein	Protein	Protein	Protein	Protein
Source	Type	Content	Source	Type	Content
hemp seed	nut, oilseed	31.6%	flax seed	oilseed	18.3 %
kidney bean	legume, pulse	25.9%	cashew	nut	17.4%
black bean	legume, pulse	24.4%	farro	cereal	15.6%
navy bean	legume, pulse	24.1%	wild rice	cereal	14.7%
red lentil	legume, pulse	23.9%	quinoa	pseudo-cereal	14.1%
pinto bean	legume, pulse	23.7%	oat	cereal	13.5%
lentil	legume, pulse	23.6%	soy bean	legume, oilseed	11.9%
green lentil	legume, pulse	23.6%	buckwheat	pseudo-cereal	11.1%
small red beans	legume, pulse	23.5%	pecan	nut	10.0%
large white beans	legume, pulse	23.4%	barley	cereal	9.9%
peanut	legume, oilseed	23.2%	corn	cereal	9.4%
large lima beans	legume, pulse	21.5%	white rice	cereal	7.0%
almond	nut	21.4%	green pea	legume, pulse	5.4%
chick pea	legume, pulse	20.5%	potato	vegetable	2.6%
sunflower seed	oilseed	18.9%	mushroom	fungus	2.4%

**Table 2 foods-13-03495-t002:** Texture profile analysis parameters stiffness, hardness, cohesiveness, springiness, resilience, and chewiness, with units, descriptions, and equations, with the variables defined in Figure 5.

Parameter	Unit	Description	Equation
**Stiffness**	MPa	slope of stress–strain curve during first compression when compressed to a preset percentage of specimen height	$E=\frac{\sigma }{\varepsilon }$
**Hardness**	N	peak force at first compression when compressed to a preset percentage of specimen height	$F_{1}$
**Cohesiveness**	–	material integrity during second loading/unloading cycle relative to first cycle, where 0 denotes complete disintegration	$\frac{A_{3}+A_{4}}{A_{1}+A_{2}}$
**Springiness**	–	material response time during second loading cycle relative to first cycle, recovery related to viscous properties	$\frac{t_{2}}{t_{1}}$
**Resilience**	–	material recovery from deformation during first unloading relative to first loading, 1 denotes elastic and >1 plastic behavior	$\frac{A_{2}}{A_{1}}$
**Chewiness**	N	material ease of biting hardness times cohesiveness times springiness	$F_{1}\frac{A_{3}+A_{4}}{A_{1}+A_{2}}\frac{t_{2}}{t_{1}}$

## Data Availability

No new data were created or analyzed in this study. Data sharing is not applicable to this article.

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
