# Peer review of "Mimicking Mechanics: A Comparison of Meat and Meat Analogs"

_foods, 2024, doi:10.3390/foods13213495_

Round 1

Reviewer 1 Report

Comments and Suggestions for Authors

The review describe the need for standardising texture analysis of meat analogs and gives examples of  different methods. Overall, the review is well-written and I only have a few comments.

Line 39 and 46-48 and further on: It is confusing that you uses italic here as it is not citations. Please change it throughout the manuscript

Line 49: milk and egg protein are animal sources!

Line 60: I would suggest you changed ‘protein product’ to ‘protein source’ and ‘protein source’ to ‘protein type’ to use the same wording as in Table 1.

I like your figure 1

Line 85-90: what are the relation between the different topics in this section? It seems out of context for me, that you first describe environmental impact of milk and egg, then talk about meat-plant relations and end talking about cultured meat. And furthermore, the relation between the overall aim and this section is not clear. I will suggest that you delete it.

Figure 2. I will suggest another layout which more clearly shows the message of the figure. I had to read the Figure label twice to be sure what you wanted to show me in the figure. And after reading the text I am uncertain why you don’t have extrusion in the center circle in the bottom, if that should lead to the right bottom circle (whole muscle)?

3. Evaluating the mechanical properties of meat

                          Do you mean …. Of meat analogues?

Line 124-130. I agree with you in the importance of understanding the eating biomechanics, and I might be mistaken, but I do not see how you use this need of knowledge later in the review?

Line 129 – I suggest changing ‘elderly people’ to ‘older adults’

Section 3.2.2 TPA. Just a reflection – does the TPA result depends on how the meat analoges have been cooked? I know that cooking influence the texture a meat to a large extend, but do not recalled that I have seen any publications on this aspects in meat analoges.

Author Response

Reviewer #1

Thank you for your comments and suggestions.

Line 39 and 46-48 and further on: It is confusing that you uses italic here as it is not citations. Please change it throughout the manuscript.

We have used italics to emphasize certain points in the manuscript.

Line 49: milk and egg protein are animal sources!

Thank you for this correction, we have changed the text to say “alternative protein sources.”

Line 60: I would suggest you changed ‘protein product’ to ‘protein source’ and ‘protein source’ to ‘protein type’ to use the same wording as in Table 1.

Thank you for this comment, we have changed the text to say “Figure 1 and Table 1 summarize 30 alternative protein sources with their protein type and protein content.”

I like your figure 1

Thank you!

Line 85-90: what are the relation between the different topics in this section? It seems out of context for me, that you first describe environmental impact of milk and egg, then talk about meat-plant relations and end talking about cultured meat. And furthermore, the relation between the overall aim and this section is not clear. I will suggest that you delete it.

We have added a topic sentence for clarity:

While plant, fungal, and algal protein sources are both environmentally friendly and suitable for a vegan diet, researchers are also looking into animal-based alternative protein sources and reduced meat formulations that have the potential to closely match the taste and texture of meat with the promise of reduced environmental impacts and improved health compared to traditional meat products (Guyomarc’h, 2021).

Figure 2. I will suggest another layout which more clearly shows the message of the figure. I had to read the Figure label twice to be sure what you wanted to show me in the figure. And after reading the text I am uncertain why you don’t have extrusion in the center circle in the bottom, if that should lead to the right bottom circle (whole muscle)?

Thank you for this comment, we have updated Figure 2 to clarify our intention of providing an outline of the protein sources, processing methods, and products involved in the creation of meat analogs that are either in commercial use or in development.

  1. Evaluating the mechanical properties of meat
    Do you mean .... Of meat analogues?

We have changed the title to:

Evaluating the mechanical properties of meat and meat analogs

Line 124-130. I agree with you in the importance of understanding the eating biomechanics, and I might be mistaken, but I do not see how you use this need of knowledge later in the review?

We have provided this very brief overview of eating biomechanics to both motivate the need for rigorous mechanical testing and to provide relevant citations to comprehensive review papers for a reader who may not be familiar with the eating biomechanics literature.

 We have added to this paragraph:

Understanding how people chew meat is essential to designing mechanical tests that capture the sensory experience of different aspects of the chewing process.

Section 3.2.2 TPA. Just a reflection – does the TPA result depends on how the meat analoges have been cooked? I know that cooking influence the texture a meat to a large extend, but do not recalled that I have seen any publications on this aspects in meat analoges.

Thank you for this reflection, it is a very interesting topic that we agree is under-explored! I have found two such studies, both comparing how cooking affects meat vs. meat analog patties.

  1. Souppez et al. Mechanical properties and texture profile analysis of beef burgers and plant-based analogues. Journal of Food Engineering, 2025.
  2. Vu, Zhou, and McClements. Impact of cooking method on properties of beef and plant-based burgers: Appearance, texture, thermal properties, and shrinkage. Journal of Agriculture and Food Research, 2022.

We have added the following to section 3.2.2.:

In the case of meat, standardizing cooking time and temperature is another important element to consider, although the effect of cooking on texture profile analysis parameters for meat analogs is not yet clear, with one study reporting equal changes in parameter values for meat analog patties as meat patties (Souppez 2025), while another study found greater changes for the meat patties (Vu 2022).

Reviewer 2 Report

Comments and Suggestions for Authors

Thank you for giving me an opportunity to revise the manuscript:

Formulation of meat analogs include proteins, polysaccharides and their interactions contribute significantly to the texture. Why the manuscript focuses only the proteins and proteins sources?

The authors really need to emphasize on the novelty of this article and deeply explain the need to investigate this topic.

I suggest the authors to modify Table and give various components of the meat analogs ( protein, polysaccharides etc).

Under section 2.2 processing methods.

can the authors discuss more on effect of different processing method on the texture of meat analogs?

One of the objectives of the manuscript was to make recommendation for improvements. However, under discussion parts this aspect was not well covered. I suggest the authors to provide more discussions about this aspect.

Author Response

Reviewer #2

Thank you for your comments and suggestions.

Formulation of meat analogs include proteins, polysaccharides and their interactions contribute significantly to the texture. Why the manuscript focuses only the proteins and proteins sources?

Thank you for this comment, we have added the following to section 2.2. We have limited our scope to protein sources since the main goal of this review is to comment on mechanical testing.

Ultimately, for both comminuted and whole muscle analogs, the final texture is the combined result of the protein source, non-protein ingredients, pH and salts, temperature, and degree of shear induced by the processing method (Beniwal 2021).

The authors really need to emphasize on the novelty of this article and deeply explain the need to investigate this topic.

Thank you for letting us clarify the novelty in a bit more detail. We have highlighted the following in the introduction:

Here we review the most popular testing methods for meat, discuss their advantages and limitations, and make recommendations for improvement, with the goal to standardize meat testing and enable rigorous cross-study comparisons.

For the first time in the literature, we then present compiled cross-study values of the Warner-Bratzler shear force and stiffness of various whole and comminuted meats and meat analogs.

I suggest the authors to modify Table and give various components of the meat analogs ( protein, polysaccharides etc).

Thank you for the suggestion, but we have decided to leave Table 1 since Reviewer #1 really liked this Table and Figure.

Under section 2.2 processing methods. can the authors discuss more on effect of different processing method on the texture of meat analogs?

We have added the following:

Additionally, extrusion outputs a dense, impermeable product that cannot both keep fiber orientation and be shaped into whole muscle cuts of a realistic size (Dikovsky 2024).

Yet, shear cell technology is also unable to provide a texture that truly mimics beef (Dikovsky 2024).

One of the objectives of the manuscript was to make recommendation for improvements. However, under discussion parts this aspect was not well covered. I suggest the authors to provide more discussions about this aspect.

Thank you for raising this point. We have added the following recommendations:

Given the wide variability of the Warner-Bratzler shear test and texture profile analysis, we propose to use the sample stiffness as the most important parameter to report.

Additionally, we recommend to prescribe the strain rate and reported it in relative units of %/s rather than absolute units of mm/s to control the total time to deform.

Although both Warner-Bratzler shear force and stiffness are simple to test and provide straight-forward comparison metrics, these values are from a single mode. When sample quantity allows, we recommend to perform multi mode testing, such as tension, compression, and shear, to probe the full 3D mechanical behavior (St. Pierre 2023, St. Pierre 2024).